# Learning to Rewrite Prompts for Personalized Text Generation

## ABSTRACT

Facilitated by large language models (LLMs), personalized text generation has become a rapidly growing research direction. Most existing studies focus on designing specialized models for a particular domain, or they require fine-tuning the LLMs to generate personalized text. We consider a typical scenario in which the large language model, which generates personalized output, is frozen and can only be accessed through APIs. Under this constraint, all one can do is to improve the input text (i.e., text prompts) sent to the LLM, a procedure that is usually done manually. In this paper, we propose a novel method to automatically revise prompts for personalized text generation. The proposed method takes the initial prompts generated by a state-of-the-art, multistage framework for personalized generation and rewrites a few critical components that summarize and synthesize the personal context. The prompt rewriter employs a training paradigm that chains together supervised learning (SL) and reinforcement learning (RL), where SL reduces the search space of RL and RL facilitates end-to-end training of the rewriter. Using datasets from three representative domains, we demonstrate that the rewritten prompts outperform both the original prompts and the prompts optimized via supervised learning or reinforcement learning alone. In-depth analysis of the rewritten prompts shows that they are not only human readable, but also able to guide manual revision of prompts when there is limited resource to employ reinforcement learning to train the prompt rewriter, or when it is costly to deploy an automatic prompt rewriter for inference.

## KEYWORDS

prompt rewrite, personalized text generation, large language models

### ACM Reference Format:
Anonymous Author(s). 2018. Learning to Rewrite Prompts for Personalized Text Generation. In *Proceedings of (WebConf '24)*. ACM, New York, NY, USA, 11 pages. https://doi.org/XXXXXXX.XXXXXXX

## 1 INTRODUCTION

With the flourishing of artificial intelligence (AI)-powered content creation, personalized text generation has become a popular research area. Applications of personalized text generation span a wide range of domains, including AI-assisted writing of various content types (e.g., tweets, news articles, scientific papers, and fictions), corporate and personal communications (e.g., emails, chats and forum posts), and the transformation of written content into

different styles (e.g., summarization or elaboration). Given its importance, it is crucial to develop generative systems that cater to specific audiences, creation contexts, and information needs.

Most prior work on personalized text generation relies on domain-specific features or knowledge and proposes models to address a particular domain, such as reviews [12, 13], dialogue agents [16, 28, 29] and social networks [5]. Recently, large language models (LLMs) have been utilized to produce personalized text across domains [11, 22], and fine-tuning is usually required to adapt the LLMs to personal data and the personalized text generation task.

When fine-tuning is feasible, previous work [11] has found that the personalized text generator produces the best results when it is instructed by a prompt that includes descriptions about the user's personal context that are processed through multiple stages. These stages include *retrieval*, *ranking*, *summarization* and *synthesis*. Specifically, given a personalized writing task, relevant entries from the author's personal history are retrieved and ranked; from the retrieved entries, important sentences are extracted as a summary and important keywords are extracted as a synthesis of the author's personal context. A prompt is then formed by concatenating the task instruction, the summary, the synthesis, and the ranked entries.

In most real-world cases, however, fine-tuning the LLM, or the content generator, is not feasible. On one hand, the LLM is often a black box with frozen parameters and one has no control over its internal mechanism. This scenario is typical when one interacts with the popular LLM applications such as ChatGPT[1] and Bard[2]. These models are capable of generating human-like text but are often only accessible through text-based interfaces or APIs. On the other hand, even if one has access to the model parameters, fine-tuning a LLM typically requires an inhibiting amount of computational resources for individual users. In either case, the only way one can influence the output of the LLM is by providing different text prompts. In this paper, we focus on this practical scenario and propose to learn personalized text prompts to achieve better generation results.

In this scenario, the prompts that are effective for fine-tuning a model might not be optimal for instructing a frozen LLM. The reasons are two fold. First, a frozen LLM is more likely to be affected by the noise in the input. For example, certain summary sentences produced in the *summarization* stage might be off-topic. Through fine-tuning, the model can learn to cope with the noisy summary and mine relevant information from it. When the LLM is frozen, it is more likely to be severely affected by this noise and produce less relevant output. Second, certain choices of the format and content of synthesis might help the LLM better understand the context of the specific user, producing results more personalized for that user without manipulating its general knowledge about the task.

Motivated by these reasons, we train a prompt rewriter that revises particular components of the original prompt of personalized text generation (that correspond to particular stages in previous work) so that it can lead to better performance of a frozen LLM.

---

[1] https://chat.openai.com
[2] https://bard.google.com

However, the gradient of the LLM is often not accessible through APIs, making the entire training pipeline non-differentiable. Furthermore, the performance of the document generator is usually measured by word overlap metrics like Bleu [19], which are also non-differentiable. To still learn the prompt in an end-to-end manner, we employ reinforcement learning (RL) to update the prompt rewriter, where the performance metric is used as the reward.

Prior research has explored RL to optimize the *instruction* component of the prompt [2, 6, 30], e.g., *"classify the sentiment of the following text"*, which is usually independent of the *task input*[3] or context. In contrast, our work aims to optimize the *input/context-dependent* components of the prompt specifically for the task of personalized text generation, in particular the summary and the synthesis of the given user's personal context. These components are specific to both the user and the ad hoc intent of the writing task and are not shared across other users or contexts.

The search space for instruction optimization is relatively small, as there is a limited vocabulary to describe a general task (e.g., sentiment classification). In contrast, the search space for optimizing input/context-dependent components in a prompt is significantly larger, as for every document that a particular user is about to write, we may need a different description about this user's intended content and personal preference. Such a tremendous search space places a critical challenge to the direct application of RL. To address this challenge, we propose a training paradigm for prompt rewriting that chains supervised learning (SL) and RL. In this paradigm, the rewriter is first adapted to the prompt rewrite task using supervised learning, followed by further optimization using RL.

To evaluate the prompt rewriter, we conduct experiments on datasets from three representative domains: personal email communications, product reviews, and social media discussions. Results demonstrate that the rewritten prompts through our approach outperform both the original prompts generated by the state-of-the-art method and the prompts optimized via SL or RL alone on all three domains. This suggests that our SL-RL paradigm is a promising approach for optimizing prompts for personalization.

In addition, we conduct an in-depth analysis of the learned prompts. These prompts are not only human-readable, they also indicate general rules that explain how the prompts are revised. Using these rules as guidance to manually revise the original prompts, we can obtain significantly better performance than the original prompts. Such guidance is particularly useful when there is limited resource for reinforcement learning to train the prompt rewriter (which requires thousands of API calls and the calculation of rewards), or when it is costly to deploy an automatic prompt rewriter.

## 2 RELATED WORK

Two research directions closely related to our work are personalized text generation and prompt learning.

*Personalized text generation.* Researchers have studied personalized generation for a specific domain by utilizing domain-specific

---

[3]In literature, the *task input* is also referred to as the *query*, *instance*, or *content* of a prompt, depending on what the input to the task actually is, and to be distinguished from the general description (i.e., the *instruction*) of the task or the few-shot *exemplars*. In our case, the task input is the immediate context of the writing task and the personal context of the user (see Section 3), so we use *context* and *input* interchangeably.

features or knowledge. For example, Li and Tuzhilin [13] use self-attentive recursive autoencoders to generate personalized user reviews given product descriptions, sentiment labels, and user historical reviews. Product attributes are utilized in [12] to improve personalized review generation. Gao et al. [5] propose to feed personalized features to the encoder to guide the decoder for personalized social text generation. There are extensive studies on personalizing dialogue agents [16, 28, 29]. Due to the lack of real data, they have explored constructing dialogue data by asking crowd-workers to write dialogues for personas [29], extracting user attributes and utterances from Reddit [16, 28] and Weibo [20, 31].

Utilizing large language models for personalized generation across different domains is a new direction. LaMP [22] provides a benchmark for training and evaluating personalized language models on classification and sentence-level generation tasks. The multi-stage and multi-task framework proposed in [11] considers the generation of passage-length personalized outputs. Skopyk et al. [26] propose to train transformer layer adapters for personalization, but experimental analysis is not included.

Our work differs from previous studies in that we consider the practical setting where LLMs, which are the foundation models for personalized text generation, are frozen and can only be accessed through APIs. In this setting, one can only improve the text prompts to enhance the generation performance.

*Prompt learning.* There have been extensive studies on automatically learning continuous prompts (a.k.a. soft prompts) in an embedding space, such as prefix tuning [14] and prompt tuning [9]. These methods require access to the parameters of the LLM (even though they are not updated) so that additional parameters can be trained. In contrast, we focus on the typical scenario where the LLM can only be accessed through APIs, and we can only modify the discrete prompts (a.k.a. hard prompts) to improve the generation. Therefore, we focus our survey on learning discrete prompts.

Researchers have investigated various approaches to assist the composition of prompts. Retrieval-augmented generation [10] generates text conditional on relevant documents, by retrieving them and including them into the prompt. Paraphrasing-based methods [7, 8] rewrite a given prompt into multiple candidate prompts and select the one with the best performance on the training set.

One line of research most relevant to ours is automatic prompt generation. Zhou et al. [32] feed example input-output pairs to a prompting LLM to generate instructions, the quality of which is evaluated by the zero-shot performance of the target LLM, which is prompted by the generated instructions. Gao et al. [4] employ the T5 models [21] to generate prompts based on the task of filling in missing spans. A gradient-based search method [25] is proposed to generate prompts for classification tasks. The work closest to ours employs reinforcement learning (RL) to generate prompts, with the reward being the generation performance of pre-trained language models [2, 6]. The learned prompts are applied to the controlled text generation task [6] and the text style transfer task [2]. The above RL based work learns input/context-independent parts (e.g., the instruction) of the prompts, which are fixed for the same task during inference time. The TEMPERA method [30] extends this line of work by learning input-dependent instructions and exemplars and applies them to text classification tasks.

Prior work mainly focuses on optimizing the *instruction* component of the prompt via RL, with the exception of TEMPERA which also permutes/swaps the in-context exemplars. Our work instead directly optimizes the more complex and *input-dependent* components for personalized text generation via unrestricted editing operations and tackles the challenge of a much larger search space. Instead of using small language models, we directly use the personalized generation performance of a *large* language model as the reward. Unlike previous findings that RL-learned prompts are ungrammatical and unreadable by humans [2], our learned prompts are not only human readable, but can also provide interpretable guidance on how to manually improve the prompts.

## 3 PROBLEM FORMULATION

### 3.1 The FtPersLlm Framework

Since our work focuses on a similar setting as the previous work of personalizing LLMs [11], we briefly introduce their work to facilitate discussion. For simplicity, we refer to their framework to fine-tune the personalized LLM proposed in [11] as FtPersLlm.

Suppose a user is writing a document, which we call the **current document**. Given the **immediate context** $x$ and the user $u$'s **personal context**, FtPersLlm aims to finish the document so that the generated document is close to the real current document as if the user had completed it. The immediate context $x$, or the *input* to the personalized writing task, is defined as the title and the start of the current document. The user's personal context is defined as documents authored by this user in the past.

FtPersLlm is a multistage framework. Using the immediate context $x$ as the query, a retriever *retrieves* relevant entries from the user's personal context, which are then *ranked* by a ranker. The ranked entries $\mathcal{E}$ are consumed by: (1) a summarization model $\mathbf{Su}(x, \mathcal{E})$ to *summarize* the retrieved entries; and (2) a synthesis model $\mathbf{Sy}(x, \mathcal{E})$ to *synthesize* the key elements and patterns in the personal context. Different approaches can be used for synthesis. In FtPersLlm, keywords are extracted at the synthesis stage. A prompt $p = (inst, x, \mathbf{Su}(x, \mathcal{E}), \mathbf{Sy}(x, \mathcal{E}), \mathcal{E})$ is generated by concatenating the instruction $inst$ (i.e., *Finish the passage in the user voice*), the immediate context $x$, the summary $\mathbf{Su}(x, \mathcal{E})$, the synthesis $\mathbf{Sy}(x, \mathcal{E})$, and the ranked entries $\mathcal{E}$. Taking this configured prompt as input, a LLM based **document generator** is **fine-tuned** against the ground-truth current document. Clearly, $\mathcal{E}$, $\mathbf{Su}(x, \mathcal{E})$, and $\mathbf{Sy}(x, \mathcal{E})$, are **dependent** on the immediate context $x$, while $inst$ is not.

We use the best configuration learned from FtPersLlm to compose our initial prompt, where the summary and the keyword synthesis are both learned to be conditioned on the immediate context via weak supervision. Specifically, the weak labels for learning the summary are created by extracting snippets from the retrieved entries that are close in semantics to snippets from the ground-truth current document. Keywords are learned in the same way except that words are extracted instead of snippets.

### 3.2 Prompt Rewriting for Frozen LLMs

Unlike FtPersLlm [11], we consider a scenario where the document generator cannot be fine-tuned but is instead a **frozen** LLM, which is a black box that accepts a text prompt as input and generates a piece of text as output. This is a typical scenario when people

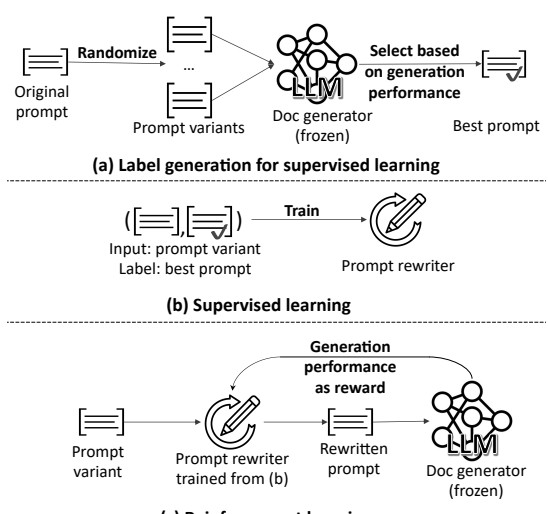

**Figure 1: The overview of our procedure to rewrite prompts for personalized text generation.**

interact with chatbots like ChatGPT[4] and Bard[5]. Given the original prompt generated by FtPersLlm, our goal is to optimize a **prompt rewriter** that rewrites the initial prompt, which in turn prompts the LLM to output a desired personalized document.

Specifically, let the training set be a list of examples $\{(p, d)\}$, where $d$ is the ground-truth current document, and the prompt $p$ consists of the instruction $inst$, the immediate context $x$, the summary $\mathbf{Su}(x, \mathcal{E})$, the synthesis $\mathbf{Sy}(x, \mathcal{E})$, and the ranked entries $\mathcal{E}$. We aim to train a personalized prompt rewriter $\mathbf{R}(\cdot)$ that rewrites the original prompt $p$ into $p' = \mathbf{R}(p)$ so that we can maximize the similarity between the generated document $d' = \mathbf{G}(\mathbf{R}(p))$ and the ground-truth document $d$, where the document generator $\mathbf{G}$ is a frozen LLM.

Without loss of generality, in this paper we only train the prompt rewriter to rewrite two critical components, the summary $\mathbf{Su}(x, \mathcal{E})$ and the synthesis $\mathbf{Sy}(x, \mathcal{E})$, while keeping other components fixed. The remaining components ($inst$ and $\mathcal{E}$) can be optimized in a similar way, or using any existing method that optimizes instructions and exemplars. We leave this as a direction for future research.

## 4 METHOD OVERVIEW

We present the overview of our procedure to rewrite prompts for personalized text generation in Figure 1.

As discussed in the introduction, optimizing the context-dependent summary and synthesis results in a significantly larger search space for reinforcement learning. To address this challenge, we perform supervised learning prior to reinforcement learning to adapt the learned model to the prompt rewriting task. As illustrated in Figure 1 (a), we first generate labels to facilitate the supervised training of the prompt rewriter $\mathbf{R}$. The labels are updated prompts that result in improved generation performance of the document generator $\mathbf{G}$. Specifically, we randomize the given original prompt $p^o$ by shuffling

---

[4]https://chat.openai.com
[5]https://bard.google.com

and removing the elements in both the summary and the synthesis components and produce a set of prompt variants $\{p_i^v\}$, including the original prompt $p^o$. We feed each variant $p_i^v$ to the document generator, a frozen LLM, to collect the generated document. We measure the generation performance against the ground-truth current document using metrics such as Bleu [19]. We refer to the prompt obtaining the highest metric score as the best prompt $p^b$.

In Figure 1 (b), we train the prompt rewriter by feeding a prompt variant $p_i^v$ as input and using the best prompt $p^b$ as the label.

Note that the label generated in step (a) is not optimal: (1) it is impractical to enumerate all prompt variants, feed them to the LLM, and find the best prompt; (2) the prompt variants can only remove or shuffle existing elements but could not introduce new but effective elements to the original prompt. Therefore we continue to train the prompt rewriter from step (b) by reinforcement learning as in Figure 1 (c). The reward is obtained by computing the generation performance using metrics like Bleu (in the same way as step (a)). The prompt rewriter $\mathbf{R}$ is then updated to maximize the reward.

## 5 LEARNING THE PROMPT REWRITER

We discuss the details of training the prompt rewriter outlined in Section 4.

### 5.1 Writing Style as Synthesis

Before describing the training details, we introduce a new type of synthesis based on writing style. Our empirical findings suggest that this method is a valuable addition to the keyword-based synthesis proposed by FtPersLlm [11].

A user's writing style is likely to remain consistent across different topics, e.g., their choice of words, preferred sentence structure, or use of grammar[6]. Therefore we synthesize a user's writing style to provide additional guidance to the document generator.

We use the same LLM used for document generation to synthesize a user's writing style based on their personal context. For simplicity and efficiency, we use the earliest documents in the user's personal history that are never used as current documents to synthesize a user's writing style regardless of the immediate context. In real applications, we can also update it periodically in case a user's writing style shifts over time.

The instruction we give to the LLM is: "*Summarize the author's writing style in detail. <Past Documents> 1.*". The number *1.* prompts the model to summarize the writing style in a numbered list.

The LLM returns phrases that describe the writing style of the user, such as *writes detailed instructions*, *professional and results-oriented*, and *well-written and engaging*. The writing style synthesis is included into the prompt in addition to the keyword synthesis (see Figure 2 (a)).

### 5.2 Label Generation for Supervised Learning

Before applying reinforcement learning, we perform supervised learning to adapt (i.e., fine-tune) a sequence-to-sequence model to the task of prompt rewriting. This process helps to reduce the search space of the RL algorithm, as discussed in the introduction.

The challenge is the lack of ground-truth labels, or optimal prompts that lead to the best generation performance. We address

[6]https://en.wikipedia.org/wiki/Writing_style

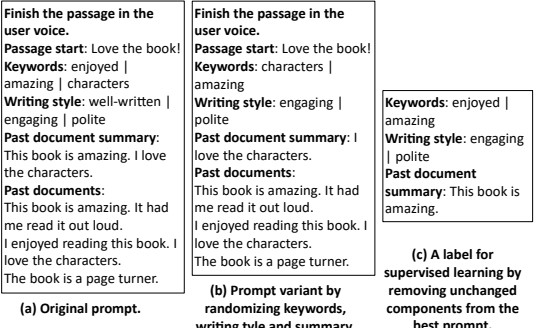

**(a) Original prompt.**

**(b) Prompt variant by randomizing keywords, writing tyle and summary.**

**(c) A label for supervised learning by removing unchanged components from the best prompt.**

**Figure 2: An example of the original prompt, a prompt variant, and the label for supervised learning. Bold parts are input-independent instructions, while plain parts are dependent on the user context.**

this by randomizing elements from the summary and synthesis, anticipating that one of the randomized versions might lead to better results and thus can be used as labels. We define three types of elements:

- Sentences in the summary.
- Keywords in the synthesis.
- Phrases describing the user's writing style in the synthesis.

For each type, we perform randomization of elements in the original prompt to produce multiple prompt variants. Our intuition is that both the selection and the ordering of elements can affect the generation performance. Specifically, we randomize the order of elements and remove the trailing $k$ elements, where $k$ is uniformly sampled from $[0, \lfloor N/2 \rfloor]$ and $N$ is the number of elements. For example, keywords $w_1, w_2, w_3, w_4, w_5$ may be permuted into a new sequence $w_3, w_2, w_5, w_1, w_4$, and then into the final sequence $w_3, w_2, w_5$ after randomly removing the last 2 words.

We produce 4 unique randomized sequences for each element type, which leads to a set of $4 \times 4 \times 4 + 1 = 65$ prompt variants $\{p_i^v\}$ per *current document*, including the original prompt. All other components (e.g., the instructions, and ranked entries) of a prompt variant remain the same as those of the original prompt.

We feed each prompt variant $p_i^v$ to the document generator and evaluate the generation performance $sc(p_i^v)$ against ground-truth current document based on metrics like Bleu. The variant with the highest score is chosen as the best prompt $p^b = \arg\max_{p_i^v} sc(p_i^v)$ and can be used as the label for supervised learning. An example of the original prompt is present in Figure 2 (a) and a prompt variant is present in Figure 2 (b).

Admittedly, there might be more sophisticated methods to generate labels other than randomization. We leave exploring better methods as future work since we find randomization to be efficient and effective: (1) it does not require enumeration of all possible permutations; (2) the best prompts improve Bleu [19] by 10 over original prompts on all datasets we experiment with.

### 5.3 Supervised Learning

A training example is created for each prompt variant as input and the best prompt as the output label. Any sequence-to-sequence

model can be trained as a prompt rewriter by minimizing the cross-entropy loss. Practically, such a model should have a smaller size than the document generator so that both training and prompt rewriting are efficient. We choose the T5-11B model [21] for most experiments since it is effective and still smaller than most state-of-the-art LLMs. We will compare models with different sizes in the experiment section.

In practice, we find that removing unchanged components from the label makes training more efficient – the prompt rewriter does not have to learn to copy the unchanged components. Hence we only keep the summary and the synthesis in the label while removing other unchanged components like instruction and ranked entries. An example of such label is shown in Figure 2 (c). Before feeding the rewritten prompt to the document generator, we add the unchanged components back to the rewritten prompt.

## 5.4 Reinforcement Learning

Though best prompts empirically show significant improvement over original prompts, using best prompts as labels for supervised learning is not optimal. First, it is almost impossible to enumerate all possible prompt variations for every current document and feed them to an expensive LLM to find the best prompt. Second, the prompt variations do not introduce new elements into the original prompt that might lead to better generated documents. As a result, the prompt rewriter learned in a supervised manner simply copies, reorders, or deletes elements in almost all cases.

To address these issues, we apply Reinforcement Learning (RL) to further fine-tune the prompt rewriter. Using the performance of the document generator as the reward, the prompt rewriter is trained in an end-to-end manner, eliminating the constraint on the quality of the generated labels. Furthermore, by including the whole vocabulary in the action space, the prompt rewriter is set free to explore advanced strategies such as adding new words.

Below we describe how we formulate our task as a RL problem.

- **Action space** consists of all tokens in the vocabulary of the prompt rewriter. This definition allows the rewriter to remove, reorder, and add new words to the original prompt.
- **State** is defined as the concatenation of the hidden states of the encoder of the rewriter and the generated tokens so far from the rewriter for a given input prompt.
- **Reward** is the performance of the document generator taking the rewritten prompt as the input, which is measured against ground-truth current documents by metrics such as Bleu. In this way, our prompt rewriter is optimized towards the document generator in an end-to-end manner.

The reward objective can be optimized by any off-the-shelf RL algorithm. We employ Proximal Policy Optimization (PPO) [23] as the RL algorithm. Note that the training examples for both supervised and reinforcement learning are selected from the same set of users, and the test set consists of a different set of users.

## 6 EXPERIMENT SETUP

We describe our experiment setup in detail in this section, including datasets, hyperparameters, competing methods, and metrics.

## 6.1 Datasets

We evaluate our models on the same set of public datasets as FtPersLlm [11]. Each of the three datasets comes from a representative domain: (1) for the Avocado Research Email Collection [18], the task is to generate personalized emails; (2) for the Amazon review data [17], the largest category *books* is used and the goal is to generate personalized reviews; (3) for the Reddit comments dataset [27], the task is to generate personalized posts or comments.

We replicate the data processing steps outlined in [11] with a few differences. First, we filter out current documents that produces less than 5 prompt variants. This is because few prompt variants means limited number of elements in the original prompts and thus we have less room to rewrite the prompts.

Second, we introduce additional knowledge into the immediate context to help the document generator understand the task. For the Amazon review data, we append the product title and the description to the original immediate context. For the Reddit data, we append the top-level post and the parent comment. For the Email data, we do not add additional content.

Third, for the Amazon review data and the Reddit data, we only include the most recent document per user as the current document instead of including all qualified documents. We add this constraint because evaluating the prompt variants per current document (see Section 5.2 for details) takes a significant amount of time when the dataset is large. We remove this constraint for the Email data since the data size is small. Note that this efficiency problem only exists at training time. Once the prompt rewriter is trained, the deployment is efficient.

Following [11], we partition the datasets by **users** so that the validation and the test sets only include documents from users that are not present in the training set. The partition ratio of users in train/validation/test sets is 85/5/10. All the performance reported in the experiment section is based on the test set.

A summary of data statistics can be found in Table 1.

## 6.2 Hyperparameters

We select the T5-11B [21] model as our prompt rewriter for most experiments. One subsection of the experiment analysis will be devoted to investigating the effect of different model sizes. In the supervised learning stage, the T5-11B model is fine-tuned using the Adafactor algorithm [24] with a base learning rate of 0.001. A linear warmup scheduler is used for the first 1,000 training steps. Additionally, the square root normalized decay of the learning rate is applied. The model is trained until its performance converges on the validation set. Decoding is performed using beam search [3] with a beam size of 4.

We use the PaLM 2 model [1], a new state-of-the-art LLM, as our document generator. It adopts temperature sampling as the decoding strategy. The parameters of PaLM 2 are frozen, and we set the temperature to 0 to make the output deterministic.

We use Bleu [19] as the metric to select the best prompt among prompt variants. It is also used as the reward to train the prompt rewriter in the RL stage.

**Table 1: Dataset statistics.**

| | #avg chars of current docs | #avg past docs per user context | #users | | | #current docs | | |
|---|---|---|---|---|---|---|---|---|
| | | | Train | Val. | Test | Train | Val. | Test |
| Avocado email | 3,648.8 | 42.3 | 501 | 27 | 53 | 13,305 | 764 | 1,227 |
| Amazon review | 918.2 | 31.0 | 342,431 | 20,121 | 39,955 | 342,431 | 20,121 | 39,955 |
| Reddit | 618.7 | 90.1 | 191,225 | 11,207 | 22,616 | 191,225 | 11,207 | 22,616 |

## 6.3 Competing methods

Prior research on prompt generation [2, 6, 30] aim at improving the **instruction** component of prompts, while we target the **input-dependent** components. Therefore, existing methods are complementary to our proposed method but are not directly comparable.

We build on top of the original prompt, which has empirically proven to be the best configuration in FTPERSLLM [11], as our baseline. The summary and the keywords for synthesis in FTPERSLLM are learned by weak supervision, as detailed in Section 3.1. We add writing style described in Section 5.1 as an additional type of synthesis. We refer to this method as ORIGINAL.

We train two variations of prompt rewriter, one using supervised learning only, denoted as REWRITERSL, and the other using RL only without the SL stage, named as REWRITERRL. We call the prompt rewriter trained via the SL-RL paradigm REWRITERSLRL.

## 6.4 Evaluation Metrics

We calculate the overlap between the document produced by the document generator and the ground-truth current document. Following [11, 13, 22], we use a variety of metrics that are widely adopted in personalized generation tasks: BLEU [19], ROUGE-1, ROUGE-2, and ROUGE-L [15], even though only BLEU is used to calculate the reward for RL.

We employ the paired t-test for statistical significance tests.

## 7 EXPERIMENTAL RESULTS

### 7.1 Overall Performance

The overall performance of all competing methods are present in Table 2. Note that the performance reported in this paper are generally lower than that in FTPERSLLM [11]. The reasons are two folds: (1) FTPERSLLM fine-tunes the document generator, which significantly boosts model performance; (2) the immediate context includes a short start of the current document, which is also the start of the ground-truth output. Fine-tuned models learn to copy the start to their output while a frozen LLM does not.

In general, REWRITERSL trained via supervised learning performs very closely to or slightly better than original prompts. REWRITERSLRL trained by RL outperforms ORIGINAL by a large margin.

*Analysis of REWRITERSL.* One possible reason that REWRITERSL does not perform well could be the poor performance of the best prompt. Therefore we calculate the performance of the best prompt on the test set and list it in Table 3. Based on this table, the best prompt performs extremely well. Note that it cannot be treated as a baseline as it is selected based on the ground-truth current document. To investigate further, we compare the original prompt, the best prompt, and the prompt generated by REWRITERSL.

**Table 2: Overall performance(%). * indicates statistically significant improvement over ORIGINAL at the level of 0.01.**

| | BLEU | ROUGE-1 | ROUGE-2 | ROUGE-L |
|---|---|---|---|---|
| Avocado email | | | | |
| ORIGINAL | 9.59 | 30.04 | 16.26 | 23.49 |
| REWRITERSL | 9.87 | 31.24* | 16.41 | 23.53 |
| REWRITERRL | 8.62 | 25.24 | 12.31 | 16.59 |
| REWRITERSLRL | **13.18*** | **33.74*** | **21.01*** | **27.04*** |
| Amazon review | | | | |
| ORIGINAL | 5.00 | 25.73 | 7.03 | 16.32 |
| REWRITERSL | 5.14 | 25.97 | 7.18 | 16.38 |
| REWRITERRL | 3.53 | 21.65 | 5.54 | 10.39 |
| REWRITERSLRL | **13.07*** | **34.12*** | **17.01*** | **25.16*** |
| Reddit | | | | |
| ORIGINAL | 13.13 | 31.92 | 17.13 | 24.79 |
| REWRITERSL | 13.22 | 32.43* | 17.31 | 24.85 |
| REWRITERRL | 9.62 | 21.83 | 11.72 | 15.34 |
| REWRITERSLRL | **20.59*** | **40.28*** | **27.68*** | **34.35*** |

**Table 3: Performance(%) of the best prompt variants. Not comparable to Table 2 as they are selected per ground-truth.**

| | BLEU | ROUGE-1 | ROUGE-2 | ROUGE-L |
|---|---|---|---|---|
| Avocado email | 20.88 | 44.95 | 28.74 | 36.47 |
| Amazon review | 15.72 | 40.58 | 18.83 | 28.72 |
| Reddit | 25.64 | 46.52 | 30.92 | 38.15 |

Multiple reasons lead to the marginal improvement of REWRITERSL. First, some elements in the best prompt, e.g., keywords or summary sentences, happen to be more relevant to the ground-truth current document. But it is hard for the supervised rewriter to learn why they are more relevant than those removed merely based on the immediate context. Second, multiple prompts might achieve very similar performance as the best prompt, meaning that there are elements in the best prompt whose order or presence do not really matter. Hence it is hard for REWRITERSL to learn general patterns that lead to better prompts. Therefore REWRITERSL only performs very few modifications and thus achieves similar performance as the original prompt. Third, the summary and the keywords from ORIGINAL are learned through weak supervision by T5-11B [21], which already achieve decent performance and it is not easy for REWRITERSL to improve them further.

*Analysis of RewriterRL.* Without supervised learning as the prior stage, RewriterRL performs poorly. The action space is too large for the RL algorithm to efficiently search for a policy with high rewards. Indeed, we observe that prompts produced by RewriterRL without pre-training contain meaningless tokens.

*Analysis of RewriterSlRl.* With supervised pre-training, RewriterSlRl is well adapted to prompt rewriting – it explores the search space more efficiently and produce human-readable prompts. Using the performance of the document generator as the reward, RewriterSlRl is trained end-to-end without depending on the quality of the generated labels or best prompts. Moreover, guided by prior knowledge from supervised learning, the prompt rewriter is free to explore strategies like adding new words because all tokens in the vocabulary are considered in the action space. The advantages of RewriterSlRl bring a significant gain over Original.

## 7.2 Case Studies and General Patterns Learned

We conduct an in-depth, qualitative analysis of the prompts rewritten by RewriterSlRl, presenting representative examples in Table 4 (and examples of generated documents in Table 7 from Appendix). Our analysis reveals several general patterns.

*Analysis of the summary.* The entire *summary* section is removed from the prompts for all datasets. We find that this is because the summary is composed of snippets extracted from retrieved historical entries [11], meaning that some parts of the summary are relevant to the current context but some are off-topic. Interestingly, FtPersLlm still finds the summary useful since the fine-tuned LLM learns to select important phrases to use from the summary, while the frozen LLM does not learn to filter from the entire summary.

*Analysis of the keyword synthesis.* We find multiple interesting patterns of how the keyword synthesis is rewritten.

First, we see two types of new keywords added to the prompt. (a) Keywords about the current topic, which could be synonyms or simply original words in the immediate context. We find that even copying the words from the immediate context as keywords could help the LLM keep to the topic. For example, adding the keyword *baseball* for a baseball book review helps the model stay on the topic about baseball. (b) Common words used frequently by the author in the retrieved entries are added, e.g., *totally* or *literally*.

Second, important keywords are repeated. In FtPersLlm, the synthesis model is trained to output unique keywords. Interestingly, RewriterSlRl learns to repeat keywords to convey their degree of importance to the document generator. As a result, the document generator either uses these repeated keywords more frequently, or composes the document around these keywords.

Third, keywords that are less relevant to the current topic are removed. Note that the synthesis model from FtPersLlm has already been trained to only output potentially relevant keywords, but RewriterSlRl learns to be even more conservative on relevance by learning through the reward function that the document generator can be easily impacted by less relevant keywords.

Lastly, keywords are reordered. The synthesis model from FtPersLlm is trained to output keywords in the decreasing order of their importance. RewriterSlRl instead reorders them based on their possible appearance order in the generated output. This makes it easier for the document generator to utilize them sequentially.

*Analysis of the writing style synthesis.* We observe different patterns of how writing styles are revised on different datasets.

Interestingly, on the Email data, almost all the users' writing styles have been updated into the same text: *the author is thorough, and they make the changes they have.* Even though the second part of the writing style is vague, the first part can be easily interpreted, which mentions the word *thorough* and provides a hint to the document generator that the output should be lengthy and detailed. Indeed, the statistics from Table 1 show that the Email data contains the longest documents of all the three datasets.

On the Amazon reviews, the entire *writing style* section is removed. We suspect that this is because topics in books reviews are relatively narrower than other datasets, and a user's writing style can already be easily inferred from retrieved past reviews. Writing style inferred in this way can also be more accurate than general descriptions generated based on random reviews of this user.

On the Reddit data, many phrases that are more *stylistic*, e.g., *sarcastic* and *logical*, are removed. RewriterSlRl prefers phrases that describe interests of users, or topics a user is going to write given the immediate context, e.g., *likes whiskey* and *enjoys music*.

## 7.3 Learned Patterns to Improve Original Prompts

We study if one can manually improve the original prompts using patterns learned by RewriterSlRl, which is valuable for users with limited resource to train or deploy a prompt rewriter. Our experiments suggest that it can be beneficial to prioritize the synthesis of the user's personal context with keywords and writing style in the prompt for new tasks. When experimenting with a new task of writing long documents, it may be beneficial to explore adding a writing style instruction that encourages the output to be *thorough*. We refer interested readers to Appendix A for more details.

## 7.4 Ablation Study

RewriterSlRl automatically learns to remove less useful element types. We further verify whether RewriterSlRl is adopting the right strategy by an ablation study. In Table 5, we remove element types one at a time from the input prompt to RewriterSlRl.

The observations are still aligned with the patterns learned by RewriterSlRl. (1) Keywords are important in all three datasets, as RewriterSlRl$_{word}$ performs better than all Original variants. (2) For the Amazon data, removing writing style or summary produces no effect on the performance, indicating that only the keywords are important. (3) For the other two datasets, removing writing style leads to performance drop while removing summary does not, suggesting that writing style is important but summary is not.

## 7.5 Impact of Rewriter Model Size

The experiments above are based on using the T5-11B model [21] as the prompt rewriter. Though T5-11B is already smaller than the document generator, or state-of-the-art LLMs, we are interested in whether one can use an even more lightweight model for rewriting.

Table 6 shows the performance of T5 models with different sizes as the rewriter. Though all these models still bring in considerable benefits of prompt rewriting, reducing the size of the rewriter model does result in a degradation of the performance.

**Table 4: Examples of rewritten prompts.**

Note: *Italic* words are included in the immediate context as part of the prompt. To save space, non-learnable components like the immediate context or the retrieved entries are omitted in the prompt examples, and the less important parts of the summary section are truncated.

**Example 1 Ground-truth doc**: *Yeah. My great old granddad passed away last September and these days, I wish I could call him up and talk about my real worries. Y'know, where my* life is going, whether I'm a good man... He was so damn wise, so gentle. He'd never use a sharp word where a kind one would do, and would go through hell for a friend. I loved him, and i miss him, and now he's gone, and I wish I'd had all these conversations with him.

**Original prompt**: Past document summary: Around about my age, we live in the same country, we've met up for drinks and a chat. I'd consider him quite a good friend. When our sister left, we went through a lot of stuff together after that. We went from fighting to talking, from talking to laughing. Instead of competing, we began to help each other...
Keywords: years | went | through | these | talk | stuff | she | real | our | old | little | life | last | he's | great | friends | few | ever | enough | each | down | best | around | age | sorry | set | read | quest
Writing style: patient, answering questions and clarifying doubts | sharing their personal experience and knowledge | enjoys history

**RewriterRl prompt**: <extra_id_0>es | punctuation style: punctuation style boldfont link bold bold bold <extra_id_12> <extra_id_12>

**RewriterSlRl prompt**: Keywords: old | granddad | I wish | talk | passed | life | days | sorry
Writing style: sharing their personal experience and knowledge

**Why RewriterSlRl prompt is better**: (a) summary is less relevant and is removed; (b) less relevant keywords are removed; (c) keywords are reordered based on their possible order of appearance in generated text; (d) non-relevant phrases describing writing style are removed.

**Example 2 Ground-truth doc**: *This is the second book in a new series, and its a continuation for call of the dragon. Drakes* writing draws you into her world and won't let you go. Youll find yourself wishing for a dragon of your own. Theres plenty of action and adventure here. The story moves at a fast pace, with a seductive slow burn romance. If you are a fan of epic fantasy, you owe it to yourself to read this new series by Jessica Drake. I received an arc for an honest review.

**Original prompt**: Past document summary: This is Jasmine Walts finest work to date. She returns to her style of writing found in the Baine Chronicles. If you enjoyed that series, with the Fenris spin offs, you are going to love the wonderful characters in call of the dragon. This is a beautiful seduction. I absolutely love the world building of this series, and the interplay of the characters...
Keywords: yourself | walts | walt | ive | dragon | call
Writing style: fluid and draws readers into her world | romantic | informative and helpful, perfect for those looking to learn new skills

**RewriterRl prompt**: <extra_id_3> | <extra_id_4> | | <extra_id_5> | | | | | | dragon <extra_id_8> of <extra_id_9>. <extra_id_26>. i can't wait to read more. <extra_id_27> wal

**RewriterSlRl prompt**: Keywords: yourself | call | dragon | dragon | adventure

**Why RewriterSlRl prompt is better**: (a) summary is less relevant and is removed; (b) the new word *adventure* is added, which is relevant to the immediate context; (c) less relevant keywords are removed; (d) the keyword *dragon* is highly topic-relevant and is repeated; (e) writing style is removed as it can be more accurately inferred from retrieved reviews.

**Table 5: Performance(%) of the ablation study by removing certain element types. * indicates statistically significant improvement over the best variant of the original prompt at the level of 0.01.**

| | Bleu | Rouge-1 | Rouge-2 | Rouge-L |
|---|---|---|---|---|
| Avocado email | | | | |
| RewriterSlRl$_{word}$ | 11.15* | 30.72 | 18.28* | 24.99* |
| RewriterSlRl$_{sum,word}$ | 11.17* | 30.98* | 19.83* | 25.37* |
| Amazon review | | | | |
| RewriterSlRl$_{word}$ | 12.93* | 34.23* | 17.09* | 25.13* |
| RewriterSlRl$_{sum,word}$ | 12.90* | 34.34* | 17.18* | 25.22* |
| Reddit | | | | |
| RewriterSlRl$_{word}$ | 19.75* | 38.62* | 26.15* | 32.08* |
| RewriterSlRl$_{sum,word}$ | 19.45* | 38.48* | 25.97* | 31.97* |

**Table 6: Performance(%) of rewriter models with different sizes trained by RL. * indicates statistically significant improvement over the best variant of the original prompt at the level of 0.01.**

| | Bleu | Rouge-1 | Rouge-2 | Rouge-L |
|---|---|---|---|---|
| Avocado email | | | | |
| T5-3B | 12.19* | 30.81 | 18.91* | 24.67* |
| T5-Large (770M) | 11.91* | 31.04* | 19.13* | 24.74* |
| Amazon review | | | | |
| T5-3B | 12.17 | 33.55 | 15.96* | 23.92 |
| T5-Large (770M) | 12.15 | 33.26 | 15.26 | 23.75 |
| Reddit | | | | |
| T5-3B | 19.07* | 37.44* | 26.20* | 32.26* |
| T5-Large (770M) | 17.11* | 35.65* | 22.34* | 29.43* |

## 8 CONCLUSION

We examine a typical situation where LLMs that are tasked for text generation can only be accessed through APIs, limiting our options to revising the text prompts to improve personalized generation. We develop a prompt rewriter by combining supervised learning and reinforcement learning, which automatically updates prompts that have been carefully designed for personalization in previous work. We evaluate the proposed method on datasets from three representative domains, showing that the rewritten prompts outperform both the original prompts and the prompts revised via supervised learning or reinforcement learning alone. In addition, we conduct an in-depth analysis of the rewritten prompts. The analysis shows that the automatically rewritten prompts are not only human-readable but also provide actionable rules. These rules can guide manual prompt improvement, especially when deploying reinforcement learning or an automatic prompt rewriter is costly.

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

# Appendices

**Table 7: Examples of document generator outputs.**

Note: *Italic* words are included in the immediate context as part of the prompt.

---

**Example 1 Ground-truth doc**: *Yeah. My great old granddad passed away last September and these days, I wish I could call him up and talk about my real worries. Y'know, where my* life is going, whether I'm a good man... He was so damn wise, so gentle. He'd never use a sharp word where a kind one would do, and would go through hell for a friend. I loved him, and i miss him, and now he's gone, and I wish I'd had all these conversations with him.

---

**Document generated by ORIGINAL prompt**: I wish I could talk to him about the girl I'm seeing, and how I'm not sure if I'm doing the right thing. I wish I could talk to him about my job, and how I'm not sure if I'm happy with it. I wish I could talk to him about my friends, and how I'm not sure if I'm close enough to them. I wish I could talk to him about my family, and how I'm not sure if I'm doing enough for them.

---

**Document generated by REWRITERSLRL prompt**: I wish I could talk to my old granddad about what I'm doing with it, what I want to do with it. I wish I could ask him for advice, for guidance. I wish I could tell him about the girl I'm seeing, about the things I'm worried about, about the things I'm excited about. I wish I could just talk to him. I wish I could talk to him about everything. He was the one person I could always talk to, and now he's gone. I miss him so much.

---

**Example 2 Ground-truth doc**: *This is the second book in a new series, and its a continuation for call of the dragon. Drakes* writing draws you into her world and won't let you go. Youll find yourself wishing for a dragon of your own. Theres plenty of action and adventure here. The story moves at a fast pace, with a seductive slow burn romance. If you are a fan of epic fantasy, you owe it to yourself to read this new series by Jessica Drake. I received an arc for an honest review.

---

**Document generated by ORIGINAL prompt**: A continuation of the story from call of the dragon, and I've enjoyed both books. Walts writing is beautiful, and the characters are well developed. I'm looking forward to reading more from her.

---

**Document generated by REWRITERSLRL prompt**: Apocalyptic world with call of the dragon. Drakes are the main characters in this series, and they are fascinating creatures. The author does a great job of describing a dragon with adventure. The story is fast-paced and exciting, and it will keep you turning the pages. I highly recommend this book to anyone who enjoys fantasy novels.

---

## A IMPLICATIONS OF THE LEARNED PATTERNS FOR IMPROVING ORIGINAL PROMPTS

Inspired by the general patterns discovered by REWRITERSLRL, we investigate whether one can manually improve the original prompts using these patterns. This approach can be particularly valuable for common users who do not have the resources for reinforcement learning, or when deploying an automatic prompt rewriter is too costly. We conduct the following experiments.

*ORIGINAL variants.* We create variants of ORIGINAL based on the observation that REWRITERSLRL removes the entire section of summary or *writing style* for some datasets. We produce all possible combinations based on the presence or absence of summary, keywords, and writing style. The results are shown in Table 8, where $\text{ORIGINAL}_\emptyset$ denotes removing all three components, $\text{ORIGINAL}_{sum}$ keeps the summary only, $\text{ORIGINAL}_{stl}$ keeps the style only, and $\text{ORIGINAL}_{word}$ keeps the keywords only.

**Table 8: Performance(%) of the original prompt variants inspired by patterns learned by REWRITERSLRL. * indicates the improvement of REWRITERSLRL over the corresponding variant is significant at the level of 0.01.**

| | BLEU | ROUGE-1 | ROUGE-2 | ROUGE-L |
|---|---|---|---|---|
| **Avocado email** | | | | |
| $\text{ORIGINAL}_\emptyset$ | 7.13* | 23.23* | 14.21* | 18.80* |
| $\text{ORIGINAL}_{sum}$ | 8.08* | 24.14* | 13.70* | 19.05* |
| $\text{ORIGINAL}_{word}$ | **10.38*** | 29.66* | 15.97* | 23.14* |
| $\text{ORIGINAL}_{stl}$ | 9.15* | 29.57* | **17.16*** | 23.90* |
| $\text{ORIGINAL}_{sum,word}$ | 8.80* | 28.49* | 14.91* | 22.13* |
| $\text{ORIGINAL}_{sum,stl}$ | 7.91* | 26.77* | 14.74* | 21.44* |
| $\text{ORIGINAL}_{word,stl}$ | 9.63* | **30.55*** | 16.21* | **23.97*** |
| **Amazon review** | | | | |
| $\text{ORIGINAL}_\emptyset$ | 4.96* | 21.22* | 7.42* | 14.93* |
| $\text{ORIGINAL}_{sum}$ | 2.99* | 17.67* | 4.78* | 11.88* |
| $\text{ORIGINAL}_{word}$ | **12.09*** | **33.39*** | **15.04*** | **23.85*** |
| $\text{ORIGINAL}_{stl}$ | 6.03* | 23.43* | 8.43* | 16.37* |
| $\text{ORIGINAL}_{sum,word}$ | 4.59* | 25.65* | 6.54* | 15.95* |
| $\text{ORIGINAL}_{sum,stl}$ | 2.64* | 18.17* | 4.30* | 11.84* |
| $\text{ORIGINAL}_{word,stl}$ | 10.29* | 31.11* | 13.11* | 21.92* |
| **Reddit** | | | | |
| $\text{ORIGINAL}_\emptyset$ | 4.07* | 14.98* | 6.24* | 11.65* |
| $\text{ORIGINAL}_{sum}$ | 4.79* | 17.27* | 6.95* | 12.84* |
| $\text{ORIGINAL}_{word}$ | 15.80* | 34.09* | **21.15*** | 28.03* |
| $\text{ORIGINAL}_{stl}$ | 10.50* | 25.90* | 13.82* | 20.58* |
| $\text{ORIGINAL}_{sum,word}$ | 11.61* | 29.79* | 15.51* | 22.89* |
| $\text{ORIGINAL}_{sum,stl}$ | 8.15* | 24.15* | 10.79* | 18.01* |
| $\text{ORIGINAL}_{word,stl}$ | **16.03*** | **34.94*** | 20.87* | **28.28*** |

We make the following observations. First, REWRITERSLRL can indeed guide the users towards the direction to improve the prompts – the results after manual revision of the prompts are consistent with patterns learned by REWRITERSLRL: (1) $\text{ORIGINAL}_{word}$ and $\text{ORIGINAL}_{word,stl}$ generally outperform other variants; (2) ORIGINAL variants with summary mostly score lower than those without summary; (3) for the Amazon data, $\text{ORIGINAL}_{word}$ achieve the best performance. These observations suggests that one can prioritize to synthesize the user's personal context with keywords and writing style in the prompt for new tasks, instead of overly summarizing the relevant entries in their history. Moreover, all the ORIGINAL variants still underperform REWRITERSLRL significantly, which means that an automatic prompt rewriter still has advantage over the simple strategy of trying different combinations when resource permits.

**Table 9: Performance(%) of Original$_{word,stl*}$ with original keywords and the uniform writing style learned by RewriterSlRl. $^*$ indicates the improvement of RewriterSlRl over Original$_{word,stl*}$ is significant at the level of 0.01.**

|  | Bleu | Rouge-1 | Rouge-2 | Rouge-L |
|---|---|---|---|---|
| Original$_{word,stl*}$ | 12.14$^*$ | 33.16$^*$ | 19.81$^*$ | 26.81 |

*A uniform writing style on the Email data.* In Section 7.2, we observe that RewriterSlRl learns a uniform writing style for the Email generation, which encourages the output of the LLM to be *thorough*. We therefore replace the writing style component of Original$_{word,stl}$ by this learned description for Email generation, which is referred to as Original$_{word,stl*}$ in Table 9. Even though its performance is still lower than RewriterSlRl, the gap becomes much smaller as there is a considerable improvement over Original$_{word,stl}$. This indicates that this uniform writing style found by RewriterSlRl plays an important role in improving the generation performance. When experimenting with a new task of writing long documents, it may be beneficial to explore adding a writing style instruction that encourages the output to be *thorough*.

