# OpenReview forum: "Learning to Rewrite Prompts for Personalized Text Generation"
_ACM.org/TheWebConf/2024/Conference — TheWebConf24 Oral_

### Official Review · Reviewer_C8G4 · 2023-11-18

**Novelty:** 7
**Technical Quality:** 7

**Review:**

- The researched task is timely and important for prompt engineering.
- The paper is very well-written and easy to follow.
- The proposed approach is technically sound.
- The experimental setting is reasonable and the experiments are conducted in a comprehensive way. The result validates the effectiveness of the proposed approach.
- I enjoyed reading this paper, and hope the approach could be open sources.

**Questions:**

- Why not try paraphrasing approaches, e.g. pegasus, for the prompt variants generation?
- How many variants per prompt are generated?
- Table 6 should also add T5-11B to better compare the performance with those smaller ones.
- It would be great to discuss the cost of both the rewriter and the document generator (since the prompt length would be an important costing factor we need to pay attention to).
- Could this approach be generalised to a prompt rewriter over multiple tasks?

**Ethics Review Description:**

No ethical concerns.

**Reviewer Confidence:**

4: The reviewer is certain that the evaluation is correct and very familiar with the relevant literature

**Scope:**

4: The work is relevant to the Web and to the track, and is of broad interest to the community

---

### Official Review · Reviewer_kL53 · 2023-11-19

**Novelty:** 5
**Technical Quality:** 5

**Review:**

### Summary
This work aims to improve the prompt for personalized text generation, in the setting of accessing the text generator only through APIs. In particular, this work follows the text generation setting of the previous work [11], which generates personalized texts with the relevant contexts of the summary and the keywords that are generated from trainable models, i.e., those generated summary and keywords are injected into the input prompt of the text generator for personalization. Then, the authors train those two trainable modules by supervised learning and reinforcement learning, in order to find the best summary and keywords, used as part of the prompt for personalized text generation. The authors evaluate the proposed method, namely RewriterSlRl, on three public benchmark datasets, showing the performance improvement against relevant baselines.

---

### Strengths
* In the context of personalized text generation, the idea of optimizing the prompt is novel.
* The proposed approach, which optimizes the input prompt for text generation via supervised learning and reinforcement learning of summary and keyword generation models, is convincing.
* The proposed RewriterSlRl significantly outperforms relevant baselines on three different datasets.
* The paper is well-written and easy to follow.

---

### Weaknesses
* The novelty of this work is somewhat limited. On the one hand, in terms of the personalization modules (i.e., using the relevant historical entries with the retrieval and reranking but also using the summary and keywords from those relevant entries) for personalized text generation, this work follows exactly the same setting of the previous work [11], while the additional component (i.e., generating the writing style and incorporating it into the prompt) seems marginal. On the other hand, in terms of prompt optimization, there are several existing works [2, 6, 30] that study this topic. Yet, I also acknowledge the contributions of exploring the existing prompt optimization approach in the setting of personalized text generation though.
* There are recent works [A, B] on personalized text generation with LLMs, which are worthwhile to discuss.
* The motivation for optimizing only the summary and keyword generation modules is not clear and can be further explained in Section 3.2. In other words, the authors can easily optimize the instruction and relevant entries augmented to LLMs as well, following the proposed approach, and it is unclear why focusing on those two summary and keyword generation modules.
* In order to find the best prompt, as explained in Section 5.2, the proposed approach requires generating 65 documents per sample. This process is very time-consuming and costly, compared to the approach - original method without prompt optimization. Also, the authors can further analyze this part (to reduce the cost) by varying the number of generated documents to find the best prompt.

---

[A] PEARL: Personalizing Large Language Model Writing Assistants with Generation-Calibrated Retrievers, 2023.

[B] Knowledge-Augmented Large Language Models for Personalized Contextual Query Suggestion, 2023.

**Questions:**

Please see the main review above. The points below are minor suggestions.
* In Section 5.4, the authors mention that the reinforcement learning strategy can alleviate the problem in supervised learning: the generation of simple copies, reorders, and deletes of elements in most cases. While the authors demonstrate this with examples in Table 4 by comparing rewritten prompts across different models, it would be beneficial to present quantitative results showing the extent to which the reinforcement learning model generates new words compared to the original input prompt and context.
* All the analysis results in the main paper are displayed on the last page, while paragraphs describing them appear before the last page. This layout makes it difficult for readers to follow the paper, as it requires frequent back-and-forth referencing.

**Reviewer Confidence:**

4: The reviewer is certain that the evaluation is correct and very familiar with the relevant literature

**Scope:**

3: The work is somewhat relevant to the Web and to the track, and is of narrow interest to a sub-community

---

### Official Review · Reviewer_jGbo · 2023-11-23

**Novelty:** 4
**Technical Quality:** 2

**Review:**

Due to the limitations of large models with fixed parameters, the generation of personalized text can only be accomplished through manual prompt adjustments. In this paper, the authors train a prompt rewriter using a combination of supervised and reinforcement learning methods. The approach involves initializing prompts using the original framework and then modifying the summarization and synthesis components.

Strength.
1. The motivation for this paper is strong. This paper wants to address the current problem of parameter-frozen large models generating personalized text by manually modifying prompts
2. The idea of this work is interesting. The authors train a prompt rewriter through supervised learning and reinforcement learning approaches. In addition, the data processing approach to generate diverse prompts by randomizing prompt components is novel.

Weakness
1. This paper uses the FtPersLlm framework as a base model but does not show the results of this paper’s proposed approach compared to FtPersLlm.
2. The experiments only selected one dataset from each of the three representative fields: Avocado email, Amazon reviews, and Reddit. It is recommended to add some more text generation task datasets to increase the diversity of the dataset, such as Amazon Electronic, Book datasets, IMDb Movie dataset.
3. The article utilized the PALM model as the generator for experiments, but it is suggested to conduct ablation experiments using other large-scale models to verify the robustness of the proposed method in this study, such as ChatGPT, GPT4, Bard, etc.
4. Experimental results in the section of the article on ablation experiments are not sufficient. The authors do not show the experimental results of removing the writing style element from the input prompt.

**Questions:**

None.

**Ethics Review Description:**

None.

**Reviewer Confidence:**

3: The reviewer is confident but not certain that the evaluation is correct

**Scope:**

4: The work is relevant to the Web and to the track, and is of broad interest to the community

---

### Official Review · Reviewer_4Hqk · 2023-11-26

**Novelty:** 5
**Technical Quality:** 5

**Review:**

This paper proposes a prompt rewriting framework for personalized generation. This method follows a multi-stage framework proposed in previous work to rewrite the summary and systhesis in their prompts. To train the prompt rewritter, the authors chains the supervised learning and reinforcement learning and achieve good performance.

**Questions:**

Questions:
1. The proposed method assumes that there are user personal history transformed as the summary and systhesis. How do the authors deal with the case that there is no user history?
2. The proposed method is to rewrite the summary and synthesis, but why learn a rewriter instead of not directly learning better summarization and synthesis models based on the final performance?
3. Does the process of label generation have a high cost? The task is to generate a document, and 65 prompts for each instance will consume a lot of time. How do you address this problem?
4. Why not compare with some prompt optimization methods scuh as APE?

Typos：
Line 73-74: When -> While

**Reviewer Confidence:**

4: The reviewer is certain that the evaluation is correct and very familiar with the relevant literature

**Scope:**

3: The work is somewhat relevant to the Web and to the track, and is of narrow interest to a sub-community

---

### Official Review · Reviewer_nrs1 · 2023-12-01

**Novelty:** 5
**Technical Quality:** 5

**Review:**

In this paper, the authors addressed the challenge of generating personalized text using large language models (LLMs) which can only be accessed via APIs, thus limiting direct modifications. The authors propose a novel method for automatically revising text prompts to enhance personalization. Their approach involves a hybrid training paradigm that combines supervised learning (SL) and reinforcement learning (RL) to create a prompt rewriter. The rewriter modifies initial prompts to better summarize and synthesize personal context. The effectiveness of this method is validated using datasets from three distinct domains, demonstrating superior performance compared to original prompts and those optimized by SL or RL alone.

Strengths

1. Innovative Approach: Combining SL and RL for prompt rewriting is a novel strategy that effectively narrows down the action space for RL, leading to more efficient and effective training of the prompt rewriter.
2. Comprehensive Evaluation: The paper presents an extensive evaluation across multiple datasets and compares the performance of the proposed method against several baselines, showing consistent improvements.
3. Practicality and Generalization: The research caters to a common scenario in LLM usage and offers a practical solution that can be generalized across different domains.
4. In-depth Analysis: There is a thorough analysis of the rewritten prompts, highlighting patterns learned and providing insights for manual prompt improvement in resource-constrained situations.

Weaknesses

1. Dependence on Initial Prompts: The method's efficacy hinges on the quality of the initial prompts. Poorly constructed initial prompts might limit the effectiveness of the rewriter.
2. Complexity and Resource Requirements: The combined use of SL and RL may introduce significant complexity and computational overhead, potentially limiting the method's accessibility for some users.
3. Limited Analysis of Negative Cases: The paper lacks a detailed discussion of instances where the method fails or performs suboptimally, which would be crucial for understanding its limitations.
4. Potential for Bias: If the initial prompts or training data contain biases, these could be perpetuated or even amplified by the rewriter.

**Questions:**

Please respond to the weakness part.

**Reviewer Confidence:**

4: The reviewer is certain that the evaluation is correct and very familiar with the relevant literature

**Scope:**

4: The work is relevant to the Web and to the track, and is of broad interest to the community

---

### Decision · Program_Chairs · 2024-01-22

**Decision:**

Accept (Oral)

**Comment:**

This paper presents a novel method for optimizing text prompts to enhance personalized text generation in LLMs. It introduces a hybrid training paradigm combining supervised learning (SL) and reinforcement learning (RL) to revise prompts, validated across diverse datasets. Reviewers generally view the work as relevant and of broad interest, with its innovative approach and comprehensive evaluation being key strengths.

 Concerns mainly include the dependence on initial prompt quality, complexity, and resource requirements. Some reviewers also express reservations about the novelty of the work, suggesting that it follows existing frameworks closely, with only marginal additions. The reviewers also suggest including comparisons with other LLMs and additional datasets.